# Pore Structure Characteristics of Foam Composite with Active Carbon

**DOI:** 10.3390/ma13184038

**Published:** 2020-09-11

**Authors:** Jungsoo Lee, Young Cheol Choi

**Affiliations:** Department of Civil and Environmental Engineering, Gachon University, Seongnam 13120, Korea; dlwjdtn7509@naver.com

**Keywords:** foam composite, active carbon, micropore volume, specific surface area, pore structure

## Abstract

Characterization of porous materials is essential for predicting and modeling their adsorption performance, strength, and durability. However, studies on the optimization of the pore structure to efficiently remove pollutants in the atmosphere by physical adsorption of construction materials have been insufficient. This study investigated the pore structure characteristics of foam composites. Porous foam composites were fabricated using foam composite with high porosity, open pores, and palm shell active carbon with micropores. The content was substituted 5%, 10%, 15%, and 20% by volume of cement. From the measured nitrogen adsorption isotherm, the pore structure of the foam composite was analyzed using the Brunauer–Emmett–Teller (BET) theory, Barrett–Joyner–Halenda (BJH) analysis, and Harkins-jura adsorption isotherms. From the analysis results, it was found that activated carbon increases the specific surface area and micropore volume of the foam composite. The specific surface area and micropore volume of the foam composite containing 15% activated carbon were 106.48 m^2^/g and 29.80 cm^3^/g, respectively, which were the highest values obtained in this study. A foam composite with a high micropore volume was found to be effective for the adsorption of air pollutants.

## 1. Introduction

Air and water pollution issues directly affect our lives and are becoming increasingly significant worldwide with every passing year. According to the National Air Pollutants Emission Service in Korea, in 2016, air pollutant emissions amounted to 1248 thousand tons of NOx, 359 thousand tons of SOx, 1024 thousand tons of volatile organic compounds (VOCs), and 795 thousand tons of CO [1]. In addition, the number of sewage treatment facilities are increasing owing to industrialization, thereby making it difficult to manage pollutant emissions and demanding countermeasures [2].

In recent years, numerous studies have been conducted on adsorbents capable of removing pollutants to improve these environmental problems [3,4,5,6]. Adsorbents for removing contaminants generally use a porous material, with various types of activated carbon being widely used. Activated carbon is a porous material, has a high specific surface area, and is an adsorbent with excellent physical and chemical stability. Activated carbon exhibits various adsorption properties depending on its micropore structure [7,8]. The most common methods for measuring or evaluating the pore structure of a porous material are mercury intrusion porosimetry (MIP) and nitrogen adsorption/desorption isotherms. The MIP measurement has the disadvantage that its accuracy can decrease if there are ink-bottle pores present and there is a possibility of pore destruction by high pressure [9]. However, it has the advantage of being able to measure a wide range of pore sizes with relatively ease [10]. Gas adsorption methods have also been used to analyze the pore structure of cement-based materials for decades [11,12]. Based on the amount of gas adsorbed, the interior surface area of the pores can be estimated using Langmuir’s monolayer-based theory or the Brunauer–Emmett–Teller (BET) theory, which is based on multilayer adsorption. In addition, the pore size distribution can be obtained using Barrett–Joyner–Halenda (BJH) analysis based on capillary condensation [11].

According to the International Union of Pure and Applied Chemistry (IUPAC), pores of adsorbents such as activated carbon are classified into micropores of 2 nm or less, mesopores of 2–50 nm, and macropores of 50 nm or more. In general, mesopores act as a pathway for gas or liquid substances to move from macropores to micropores. Furthermore, macropores have been found to affect the diffusion rate [13]. The adsorption performance of activated carbon is greatly affected by the pore volume, size distribution, and specific surface area of the activated carbon. In addition, activated carbon with numerous micropores has a high adsorption capacity against low-molecular-weight pollutants [14,15,16].

Numerous studies on activated carbon have primarily focused on the analyses of the correlation between the surface area and the amount adsorbed, pore volume, and the amount of adsorbed pores formed on and inside the activated carbon [17,18,19]. Horgnies et al. [20] evaluated the reduction performance of NO_2_ and CO_2_ gas for hardened cement paste containing activated carbon. Based on the analysis results, it was confirmed that the NO_2_ and CO_2_ gas concentrations were reduced by reacting with activated carbon [20]. Recently, foam concrete with a high connection pore and specific surface area has been receiving a significant amount of attention for higher adsorption performance. Sun et al. [21] performed scanning electron microscopy (SEM) image analysis to analyze the pore structure of foam concrete based on the type of foaming agent. They found that the foam concrete with synthetic surfactant-based foam agents had a smaller pore size distribution and fewer connecting pores than that with plant and animal surfactant-based foaming agents. Kunhananda Nambiar et al. [22] investigated the correlation between the pore distribution and compressive strength of foamed concrete using optical microscope image analysis and found that the pore structure of the foam concrete greatly influenced its strength and density, and that the size of the pores and strength were inversely proportional.

Many researchers have focused on improving adsorption performance by controlling the pore size and distribution contained in activated carbon. However, there are few studies showing construction materials that efficiently remove pollutants in the atmosphere by physical adsorption using activated carbon. In this study, a foam composite containing activated carbon was fabricated to improve the adsorption performance of air pollutants by increasing the number of micropores. Activated carbon was substituted 5%, 10%, 15%, and 20% by volume of cement. The pore structures of the foam composite of each variable were analyzed via optical microscope image analysis and gas adsorption tests.

## 2. Experimental Program

### 2.1. Materials

In this study, porous foam composites were prepared to estimate the pore structure characteristics that have a significant effect on the pollutant adsorption capacity. Ordinary Portland cement (OPC) and palm-based activated carbon were used as raw materials. The densities of OPC and active carbon were 3.13 g/cm^3^ and 1.90 g/cm^3^, respectively. Table 1 shows the chemical oxide composition of OPC (obtained by X-ray fluorescence (XRF) analysis, (ZSX Primus II, Rigaku, Tokyo, Japan). The calculated Bogue phase compositions of the OPC were 53.6% C_3_S, 18.4% C_2_S, 7.0% C_3_A, and 10.0% C_4_AF by mass.

Figure 1 shows the SEM image of the palm-based activated carbon. As shown in Figure 1, the surface morphology of activated carbon has uneven cavities and fine pores. Activated carbon adsorbs pollutants as well as moisture through pores. In particular, activated carbon, which contains many micropores, has the advantage of removing gaseous pollutants.

Figure 2 shows the particle size distribution of OPC and activated carbon obtained via laser diffraction analysis. The particle size distributions of OPC and activated carbon showed a monomodal distribution with peak values at 5.87 μm and 11.57 μm, respectively. The average sizes of OPC and activated carbon were 5.42 μm and 10.20 μm, respectively.

### 2.2. Mix Proportions

The porous foam composites were fabricated using the pre-foaming method in accordance with the mix proportions as presented in Table 2. The main variable is the substitution ratio of activated carbon to cement. Foam composites were prepared by substituting 0, 5%, 10%, 15%, and 20% of the cement volume with activated carbon. The foaming agent for the production of foam composites was F-200, which was produced in South Korea, the main component of which is a peptide compound that is a natural polymer material. The density of the foaming agent was 1.19 g/cm^3^. The foaming ratio, water-to-binder ratio, and binder content of foam composites were fixed at 69%, 0.3, and 400 kg/m^2^, respectively. Foam composites of different mixtures, according to Table 2, were placed in a cylinder mold of Ø100 × 200 mm^3^. The specimens were cured in a chamber at constant temperature and humidity chamber (temperature: 20 °C ± 1 °C, relative humidity: 60 ± 3%) for 24 h. Then, the mold was demolded and stored in a chamber at constant temperature and humidity under the same conditions. Subsequently, the specimens were de-molded and stored in a chamber, again under the same conditions.

### 2.3. Test Methods

The particle size distributions of the raw materials were analyzed using Horbia’s LA-950 equipment (HORIBA, Kyoto, Japan), and the surface morphology of activated carbon was analyzed by SEM (Signal 500 by Carl Zeiss, Oberkochen, Germany). A platinum coating was applied to the top surface of the specimen.

The macropore properties of the foam composite were analyzed using the ASTM C 457 test method. To analyze the macropore characteristics, test specimens with dimensions 50 × 50 × 20 mm^3^ were prepared. After converting the optical microscope image of the foam composite into a gray-level image, the pores and matrices were prominent in black and white, simplifying the pore size measurement. The porosity was calculated by dividing the sum of the measured pore areas by the total area. To analyze the specific surface area and micropore characteristics of the foam composite, BET analysis was performed using the ASAP 2020 (Micromeritics, Norcross, GA, USA) equipment. The samples were degassed at a temperature of 573 K for 24 h under vacuum before analysis to remove moisture and adsorbed contaminants on the surface. After the degassing process, the adsorption isotherm was acquired by obtaining the relative pressure and the amount of adsorption at 77 K.

The measurement range of the adsorption isotherm was between 0 and 1, and the specific surface area was measured by applying the BET theory [23] to the measured adsorption isotherm. When measuring the specific surface area, N_2_ gas was used as the adsorbent, and the cross-sectional area occupied by one molecule of N_2_ was calculated to be 0.162 nm^2^. The micropore volume and micropore surface area of the foam composite were obtained by the t-plot method [24,25] using the Harkins and Jura thickness equation of the following Equation (1). The Harkins and Jura thickness curve equation was derived by fitting the data for various metal oxides.
(1)$$t=\sqrt{\frac{13.99}{0.034-\log_{10}\frac{P}{P_{0}}}}$$

Here, *t* is the film thickness (Å) of nitrogen adsorbed on the sample surface, and *P* and *P*_0_ are the absolute vapor pressure and saturation vapor pressure, respectively.

Figure 3 shows the results of the micropore volume and micropore surface area of the activated carbon used in this study. The micropore volume was calculated using the y-axis intercept value of a straight line obtained by linear regression of data in the range of 0.5–1 nm adsorption thickness. The micropore surface area was calculated using the slope of a straight line obtained by linear regression of the adsorption data of micropores with an adsorption thickness of 0.5 nm or less. The microporous volume and microporous surface area of activated carbon were 356.31 cm^3^/g and 468.78m^2^/g, respectively.

The adsorption average pore diameter of the foam composite was calculated using the BJH method [11].

## 3. Results and Discussion

In this study, a new concept of a porous foam composite was developed to improve the performance of air pollutant adsorption. As illustrated in Figure 4, based on a foam composite with open pores and high porosity, activated carbon, having a large number of micropores for the adsorption of air pollutants, is dispersed throughout the pore surface of the foam composite to improve the adsorption performance. This concept of porous foam composite is expected to be used in various construction materials such as sidewalk blocks, soundproof panels, parking panels, and ceiling materials.

### 3.1. Pore Structures of Foam Composite

In order to analyze the macropores of the foam composite, the surface was polished with fine sandpaper, and optical microscope images of the foam composite cross section were taken. The optical microscope images were converted into gray-level images, as shown in Figure 5. The pore size was measured by distinguishing the pores and cement matrices in black and white—the pore and cement matrices were classified as black and white, respectively. The total pore content and average pore size were calculated by summing all the measured pores. The total pore content was calculated by dividing the sum of the areas of all measured pores by the total image area. In addition, the average pore size was calculated by assuming the shape of the pores to be a circle and calculating the total number of pores.

Figure 6 shows the pore content according to the pore size of the foam composite for each variable. To calculate the pore content of the foam composite, 121 images were used for each variable. To calculate the pore content, 11 × 11 images were measured by dividing the cross section of each variable equally. Figure 5 shows the optical microscope image measured for each variable and the corresponding converted gray image sample. As shown in Figure 6, more than 95% of the pore size of the foam composite was less than 1 mm, and it exhibited a similar pore size distribution regardless of the amount of activated carbon.

The total pore content and average pore size of Plain were 32.8% and 0.277 mm, respectively. The total pore content of AC5 mixed with 5% activated carbon was approximately 3.1% larger than that of Plain, but the average pore size was 0.028 mm smaller than that of Plain. It seems that the mixing of activated carbon prevented air bubbles from becoming larger and evenly dispersed in the foam composite. Furthermore, it seems that the activated carbon in the foam composite prevents air bubbles from becoming larger owing to the air bubbles being combined, allowing them to be evenly distributed. The total pore contents of AC10, AC15, and AC20 were 34.3%, 31.2%, and 26.1%, respectively, and the total pore content tended to decrease as the active carbon content increased. The average pore sizes of AC10, AC15, and AC20 were 0.224, 0.208, and 0.224 mm, respectively. Excluding AC20, the average pore size tended to decrease as the content of activated carbon increased. It seems that the incorporation of an appropriate amount of activated carbon smooths the distribution of air bubbles.

### 3.2. Specific Surface Area of Foam Composite

Figure 7 shows the nitrogen adsorption isotherm of the foam composite according to the relative pressure. In the IUPAC, graphs expressed according to adsorption characteristics are classified from type I to type V according to their shape [26]—the nitrogen adsorption isotherm in Figure 7 corresponds to a type II curve. This type of adsorption isotherm appears mainly in porous solids with micropores, and it can be seen that an inflection point appears when the pressure is low. The inflection point indicates the formation of monolayer adsorption. It was confirmed that the inflection point occurred in the range of relative pressure less than 0.1 in all the foam composites, as can be observed in Figure 7.

For all the variables, there was an insignificant increase in adsorption, and a flat isotherm appeared in the range of 0.2 to 0.8 relative pressure. From these results, it was concluded that all the foam composites had fewer mesopores [27,28].

Figure 8 shows the results of the regression analysis for the BET model equation [23] from the nitrogen adsorption experiment results in Figure 7. The BET model is an extension of Langmuir theory, which is a theory for monolayer molecular adsorption and multi-layer adsorption.
(2)$$\frac{1}{Q\left(\frac{P_{0}}{P}-1\right)}=\frac{C-1}{V_{m}}\times \frac{P_{0}}{P}+\frac{1}{V_{m}C}$$

Here, P and $P_{0}$ represent the absolute pressure and saturation pressure, respectively, and Q represents the amount of gas adsorbed on the specimen. $V_{m}$ and C are the amount of gas absorbed on the monolayer and the BET constant, respectively. The BET constant is a material-dependent dimensionless constant and usually has a value of 50 to 200.

$V_{m}$ was calculated using the slope and y-axis intercept corresponding to each variable by regression analysis, as shown in Figure 8. The specific surface area of the foam composite was calculated using $V_{m}$ regression analysis. The equation used to calculate the specific surface area is as follows [29]:(3)$$S_{BET}\;\left(m^{2}/g\right)=\;4.355\;\cdot \;V_{m}\left(cm^{3}/g\right)$$

The values of $V_{m}$ of Plain, AC5, AC10, AC15, and AC20 calculated by regression analysis were 2.48, 8.92, 14.41, 24.45, and 19.27 cm^3^/g, respectively. The specific surface areas of Plain, AC5, AC10, AC15, and AC20 calculated by Equation (3) from the results of $V_{m}$ were 10.75, 38.85, 62.76, 106.48, and 83.92 m^2^/g, respectively.

Figure 9 shows the measured specific surface area and average pore size of the foam composite according to the replacement level of activated carbon. The specific surface area and average pore size of Plain were 10.82 m^2^/g and 10.64 nm, respectively. The value of the specific surface area of Plain is higher than that of general cement paste. According to Odler’s research results [30], the specific surface area of cement paste at 1 year old with a water-cement ratio of 0.3 to 0.5 ranged from 4.4 to 9.5 m^2^/g. The specific surface area of AC5 was 38.85 m^2^/g, which was approximately 4 times larger than that of Plain, but the average pore size was 8.55 nm, which was 2.09 nm smaller than the average pore size of Plain.

The specific surface areas of AC10, AC15, and AC20 were measured to be 62.75, 106.48, and 83.91 m^2^/g, respectively. This was approximately 6 to 10 times larger than the specific surface area of Plain. This seems to be a result of the high specific surface area of the activated carbon itself, and similar results were also found in other studies [31,32]. The average pore sizes of AC10, AC15, and AC20 were 7.17, 4.87, and 5.58 nm, respectively. The replacement level of activated carbon was up to 15%, as the amount of activated carbon increased, the specific surface area increased, and the average pore size tended to decrease. In the case of AC20, the dispersion of activated carbon was not uniform compared to AC15 when the foam composite was blended, so the specific surface area decreased and the average pore size increased compared to AC15.

As shown in Figure 9, it was found that the specific surface area and average pore size of the foam composite containing activated carbon had an inverse relationship. It was confirmed that the foam composite had a relatively large specific surface area when the average pore size was small. This appears to have increased the specific surface area due to the large distribution of small pores on the surface of the foam composite. Meltem Asiltürk et al. [32] reported that the specific surface area increased as the activated carbon content increased, but the average pore size did not decrease significantly. Excluding AC20, as the amount of activated carbon increased, the specific surface area of the foam composite increased, but the average pore size of the foam composite tended to decrease.

### 3.3. Micropores of Foam Composite

The volume and surface area of the micropores of the foam composite were obtained from the measured adsorption isotherms by means of the t-plot method using the Harkins and Jura thickness equation [24,25]. Figure 10 shows the amount of nitrogen adsorbed on the foam composite according to the adsorption thickness.

All foam composites showed a narrow adsorption distribution at a thickness of 0.5 nm or less, and the amount of nitrogen adsorbed was large at a thickness of 1.5 nm or more, as shown in Figure 10. In addition, it was confirmed that the higher the replacement level of activated carbon, the greater the amount of nitrogen adsorption of the foam composite, which seems to be due to the micropore characteristics of the activated carbon itself.

It was confirmed that the foam composites containing 5%, 10%, 15%, and 20% of activated carbon—all except for Plain—had an inflection point in the range of 0.5 nm or less in thickness. It seems that AC10, AC15, and AC20 had a monolayer adsorption at a thickness of 0.5 nm or less.

Figure 11 shows the relationship between the micropore volumes and the surface areas of the foam composites according to the activated carbon content. In the case of Plain, the micropore volume and surface area were 1.06 cm^3^/g and 7.04 m^2^/g, respectively. The micropore volume and surface area of the foam composite containing 5% activated carbon were 7.40 cm^3^/g and 18.67 m^2^/g, respectively. Compared to Plain, the micropore volume was seven times and the micropore surface area was three times larger, and it seems that the incorporation of activated carbon developed the microporous structure of the foam composite.

The micropore volumes of AC10, AC15, and AC20 were 14.28, 29.80, and 21.56 cm^3^/g, respectively, which are about 14 to 29 times larger than that of Plain. However, the increase in the micropore surface area of the specimen containing activated carbon compared to Plain did not increase significantly compared to the volume of the micropores. The micropore surface areas of AC10, AC15, and AC20 were 29.01, 69.16, and 49.00 m^2^/g, respectively, which increased by roughly 4 to 10 times compared to Plain. As shown in Figure 11, it was confirmed that the micropore volume and surface area of the foam composites, except for AC20, increased as the replacement level of activated carbon increased. It seems that the incorporation of an appropriate amount of activated carbon affects the development of the micropore structure.

### 3.4. Pore Size Distribution of Foam Composite

Figure 12 shows the dV/dlog(D) pore volume and cumulative pore volume according to the pore diameter of the foam composite. Foam composites mixed with activated carbon, AC5, AC10, AC15, and AC20 all show a narrow pore size distribution below 3 nm (see Figure 12a). The cumulative pore volume and average pore diameter of Plain were 0.028 cm^3^/g and 9.99 nm, respectively. The cumulative pore volume of AC5 was 0.078 cm^3^/g, which was 2.8 times larger than that of Plain. This is due to the micropore properties contained in the activated carbon itself (see Figure 3). The average pore diameter of AC5 was 11.10 nm, which is an increase of 1.12 nm over Plain. The cumulative pore volumes of AC10 and AC15 were 0.100 and 0.123 cm^3^/g, respectively, and showed a tendency to increase as the replacement level of active carbon increased. In foam composites, a large amount of activated carbon is not well dispersed, and agglomeration occurs. Therefore, it seems that it had a negative effect on the micropore structure.

This trend was similar for the average pore diameter. The average pore diameters of AC10, AC15, and AC20 were 9.24, 5.23, and 6.79 nm, respectively, which decreased in size up to 15% replacement level of activated carbon, but increased by 20%. This result is similar to the micropore volume and surface area analysis results of the foam composite.

## 4. Conclusions

In this study, a porous foam composite was fabricated using palm-based activated carbon with micropores in a foam composite with a large number of pores and open pores to improve the pollutant adsorption performance. The replacement level of activated carbon (5%, 10%, 15%, and 20%) to the volume of cement was used as the main variable, and the pore structures of the foam composites were analyzed. The research results obtained in this study can be summarized as follows.

Image analyses results show that the pore sizes of the foam composites were more than 95% and less than 1mm, thereby showing a similar pore size distribution regardless of the replacement level of activated carbon. Excluding AC20, the total pore content and average pore size tended to decrease as the replacement level of activated carbon increased. This appeared to be effective in dispersing air bubbles by mixing an appropriate amount of activated carbon.

As a result of the specific surface areas of the foam composites, the specific surface areas of the foam composites containing activated carbon increased by 4 to 10 times that of Plain. On the contrary, the average pore sizes of the foam composites containing activated carbon were lower than those of Plain. The higher the specific surface area of the foam composite, the smaller the average pore size. The specific surface area of the foam composite containing 15% activated carbon was the highest at 106.48 m^2^/g, which appeared to have a reaction area capable of adsorbing contaminants larger than the other foam composites.

Excluding AC20, the micropore volume and surface area tended to increase as the replacement level of activated carbon increased. In addition, it was confirmed that the higher the micropore volume of the foam composite, the greater the specific surface area. The micropore volume and surface area of AC15 were the highest at 29.80 cm^3^/g and 69.16 m^2^/g, respectively. It is expected that AC15 will have a high adsorption performance for gaseous pollutants owing to its large pores.

All of the activated carbon-incorporated foam composites showed a narrow pore size distribution below 3 nm due to the micropore properties contained in the activated carbon itself. The cumulative pore volumes of the foam composite containing activated carbon were 2.8 to 4.4 times larger than that of Plain, but the average pore diameters were all lower than that of Plain.

Foam composite with AC has a high porosity and micropores, so it is considered to be effective in adsorbing gaseous pollutants such as NOx in the atmosphere. Therefore, the foam composite with AC developed in this research can be used for sidewalk blocks, soundproof panels, etc., which is thought to help improve the air quality.

## Figures and Tables

**Figure 1 materials-13-04038-f001:**
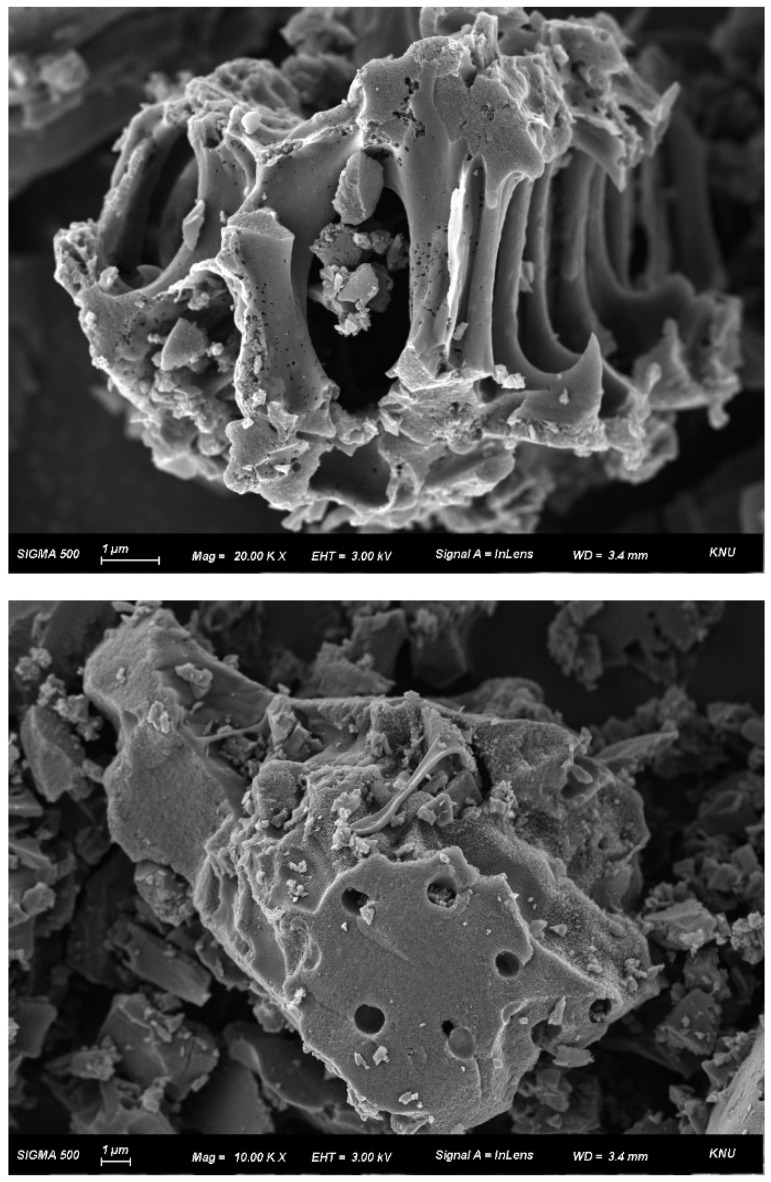
SEM image of palm-based activated carbon.

**Figure 2 materials-13-04038-f002:**
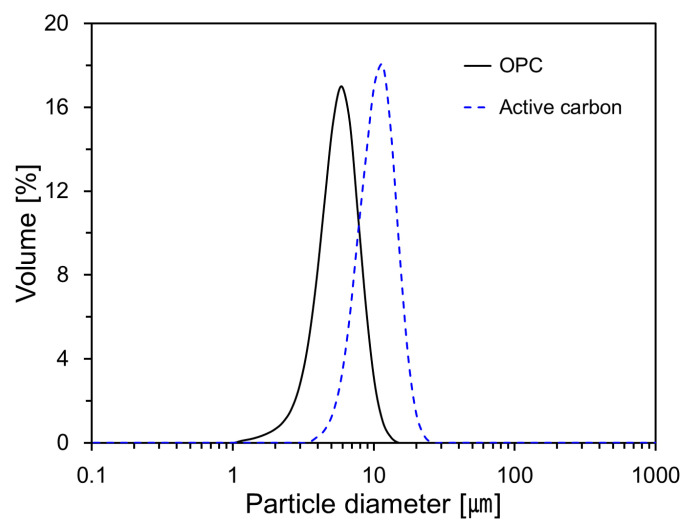
Particle size distributions of activated carbon and OPC.

**Figure 3 materials-13-04038-f003:**
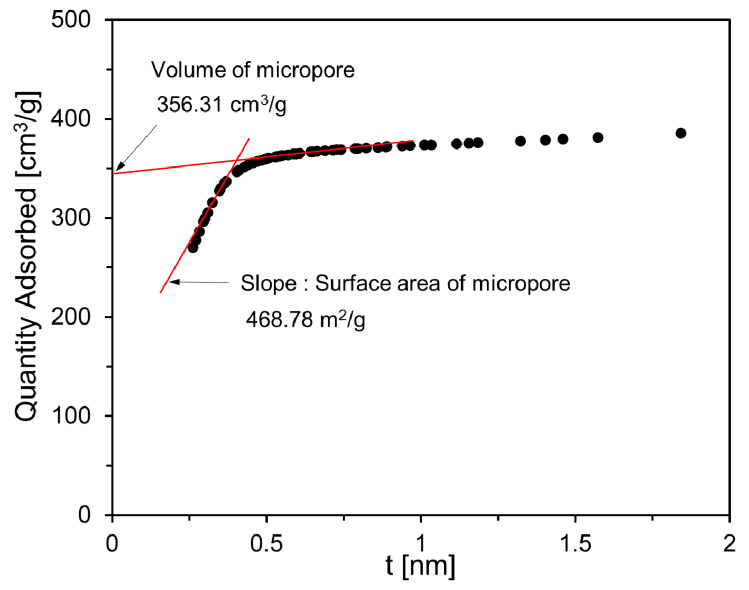
Micropore volume and micropore surface area of activated carbon.

**Figure 4 materials-13-04038-f004:**
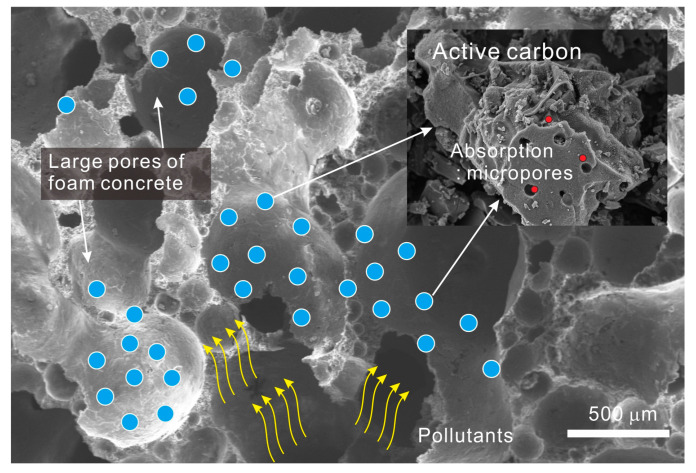
The concept of porous foam composite.

**Figure 5 materials-13-04038-f005:**
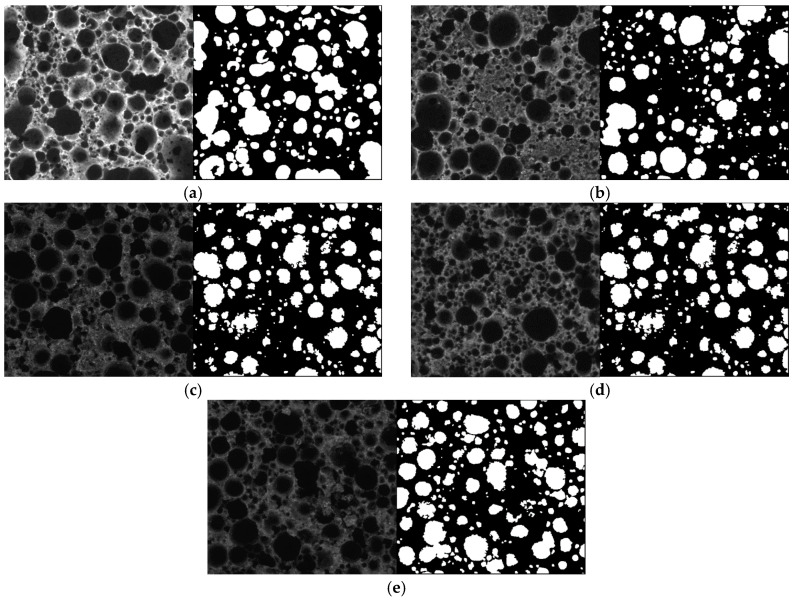
Optical microscope image analysis for porosity measurement of foam composite. (**a**) Plain; (**b**) AC5; (**c**) AC10; (**d**) AC15; (**e**) AC20.

**Figure 6 materials-13-04038-f006:**
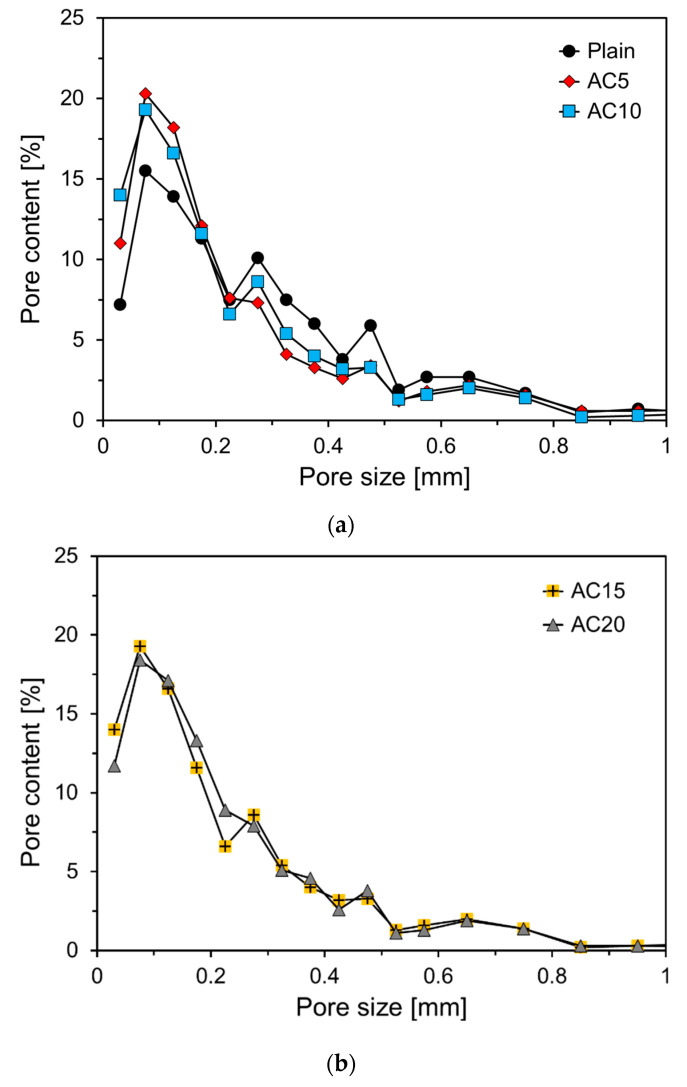
Pore contents of foam composite specimens. (**a**) Plain, AC5, AC10; (**b**) AC15 and AC20.

**Figure 7 materials-13-04038-f007:**
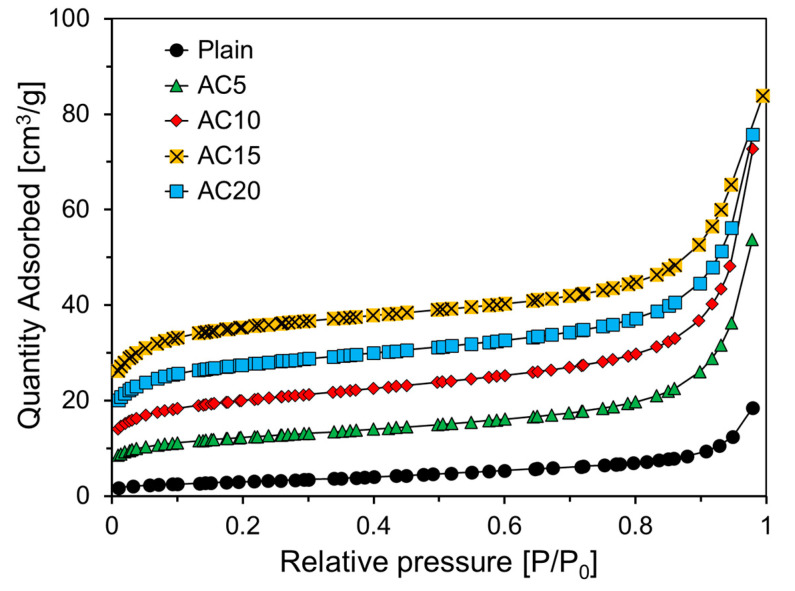
Nitrogen adsorption isotherm curves of foam composites.

**Figure 8 materials-13-04038-f008:**
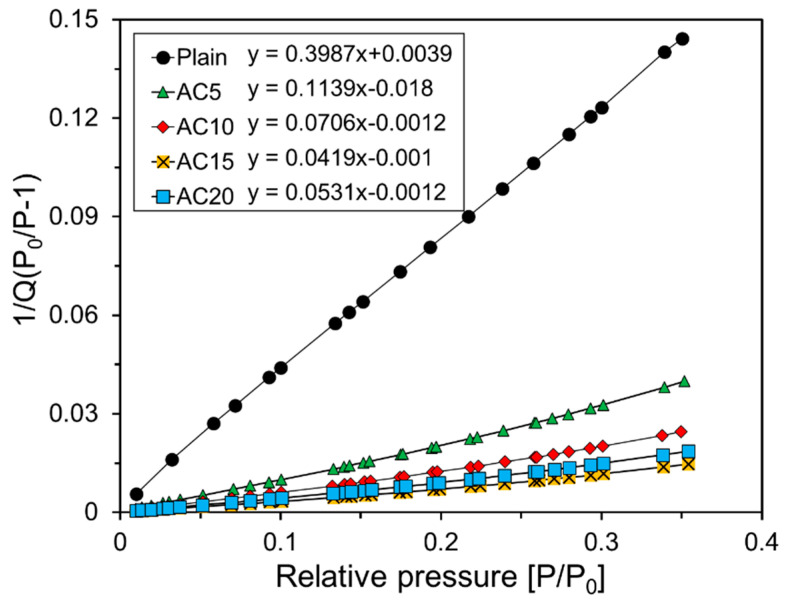
Specific surface areas of foam composites.

**Figure 9 materials-13-04038-f009:**
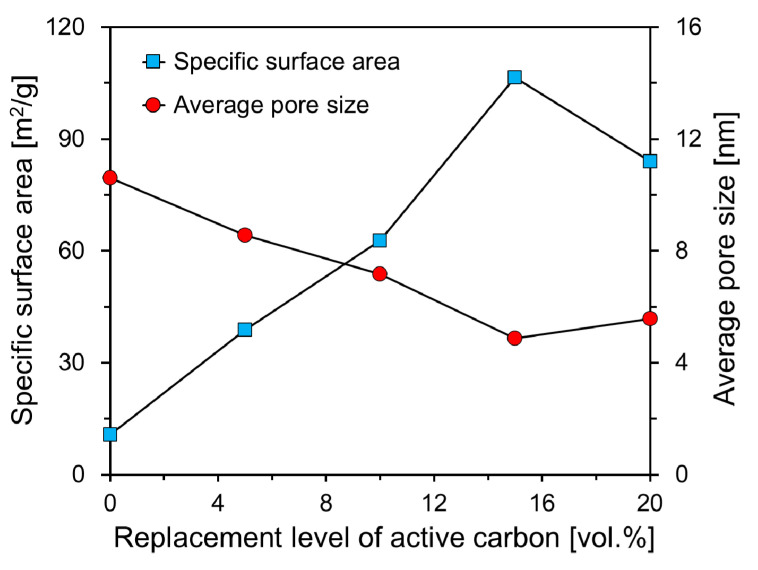
Specific surface areas and average pore sizes of foam composites according to replacement level of active carbon.

**Figure 10 materials-13-04038-f010:**
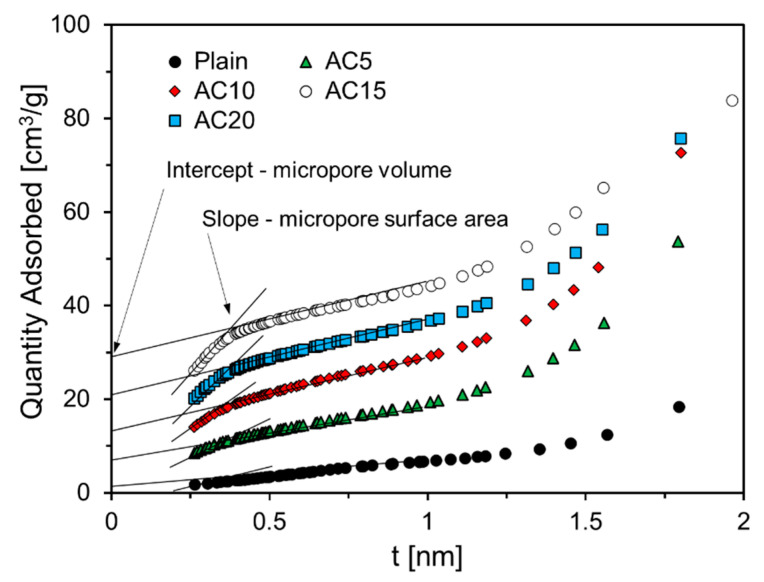
Relationship between quantity and film thickness adsorbed nitrogen of specimens.

**Figure 11 materials-13-04038-f011:**
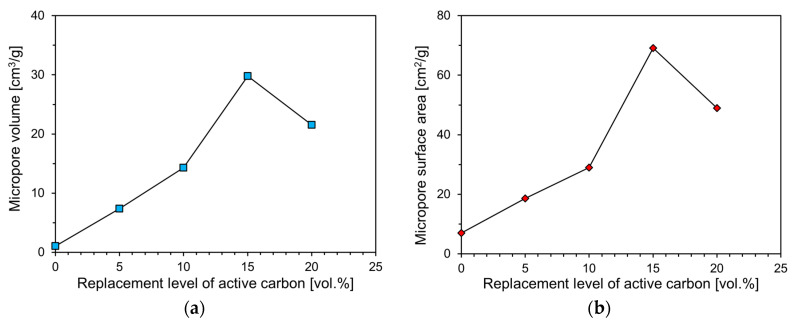
Micropore volume and surface area of specimens according to active carbon content. (**a**) Micropore volume; (**b**) Micropore surface area.

**Figure 12 materials-13-04038-f012:**
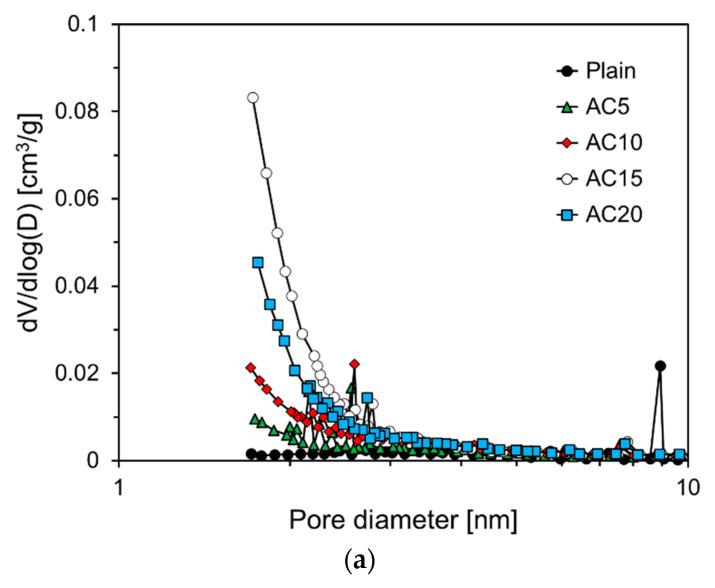
Pore size distributions and cumulative pore volume of specimens. (**a**) dV/log(D); (**b**) Cumulative pore volume.

**Table 1 materials-13-04038-t001:** Chemical oxide composition of Ordinary Portland cement (OPC) by XRF analysis.

	Chemical Compositions (wt.%)							
	CaO	SiO_2_	Al_2_O_3_	Fe_2_O_3_	MgO	K_2_O	Na_2_O	SO_3_
OPC	61.5	20.5	4.75	3.3	3.06	1.69	0.171	1.52

**Table 2 materials-13-04038-t002:** Mix proportions of foam composite.

	Target Density (kg/m^3^)	Composition of Mixture (per m^3^)				Slurry Density (kg/m^3^)
		OPC (kg)	AC (kg)	Water (kg)	Foam (m^3^)	
Plain	580.0	400.0	-	120	0.69	580.2
AC5	580.0	380.0	12.1	120	0.69	582.7
AC10	580.0	360.0	24.3	120	0.69	589.5
AC15	580.0	340.0	26.4	120	0.69	582.6
AC20	580.0	320.0	48.6	120	0.69	580.1

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
