# Peer review of "Pore Structure Characteristics of Foam Composite with Active Carbon"

_materials, 2020, doi:10.3390/ma13184038_

Round 1

Reviewer 1 Report

The authors present an interesting work on Foam Composite with Active Carbon. The work is well organized and presents interesting results. In my opinion, the paper will be able to be considered for publication after some essential revisions. Here are some comments:

Reading both the abstract and the introduction, it is not clear what is the level of innovation or the contribution of this paper to the current state of knowledge on the topic. This information should be presented more clearly in the initial part of the paper.

In subchapter 2.2, table 2 could be a little more complete and better organized and with more useful information for those who read the paper (it could be more facilitating for the reader), it could for example have the quantities of all components with the same units (may also contain the respective values in%).

Regarding chapter 3, the authors carefully present the results obtained, describing all of them in detail. However, in my opinion, something essential is missing in this chapter, that is, the critical analysis of the results obtained is missing. Authors must adequately justify the results and trends obtained, they must compare the results with each other (cross information whenever possible) and very importantly, they must compare the results obtained with values of other authors and / or with reference values. Benchmarking is essential in this type of work.

The conclusions must be reviewed. The conclusions cannot be simply a summary of the results, they must be more assertive and present conclusions more directed to the resolution of a specific problem. Things like for example: it serves ... / it does not serve ...; it is better than ... / it is worse than ...; good for this application…; not suitable for ...; etc.

Reviewer 2 Report

This paper addressed the pore structure characteristics of foam composites with activated carbon. The authors have performed some tests for comparison. However, this paper in its current state has some question should be addressed:

  1. In abstract, it should briefly describe the percentage of the activated carbon and test results such as how many percentages of the activated carbon are the best. In my suggestion, it needs to rewrite.
  2. Foam concrete was described in the abstract. However, in Table 2 did not see any aggregate.
  3. In line 272, “Plain … significantly compared to…”. It should run ANOVA to see not just the test data.

Round 2

Reviewer 1 Report

The authors answered all the questions satisfactorily and altered the paper accordingly. Therefore, I believe the paper can be considered for publication.